# Impurity effect and vortex cluster phase in mesoscopic type-1.5 superconductors

**Tian-Yi Han[1], Guo Wang[1], Jie Li[1] and Hai Huang[2]⋆**

1 School of Nuclear Science and Engineering, North China Electric Power University, Beijing, 102206, P. R. China
2 Department of Mathematics and Physics, North China Electric Power University, Beijing, 102206, P. R. China

⋆ huanghai@ncepu.edu.cn

## Abstract

Based on two-band time-dependent Ginzburg-Landau theory, we study the electromagnetic properties of two-band mesoscopic superconductors. We perform the numerical simulations with the finite element method, and determine the minimum sample size $L_c$ for the existence of the type-1.5 superconductivity from the obtained phase diagram in the absence of impurity. Meanwhile in the presence of an isotropic impurity, our numerical results reveal that the vortex cluster state induced by the attractive defect potential will gradually appear in the mesoscopic system with the sample size $L < L_c$, and the critical defect strength is about 0.2 in the $T_c$ disorder model. In addition, we also investigate the effect of anisotropic defect structures and multiple correlated disorders on the patterns of magnetic vortex distributions. Our theoretical study thus indicates that the diversity of impurity depositions has a significant influence on the semi-Meissner state in mesoscopic type-1.5 superconductors.

# 1  Introduction

Over the past two decades, two-band superconductivity has become an important research subject in condensed matter physics. This field started from the discovery of superconductivity in $MgB_2$ [1], where the existence of two distinct superconducting gaps reveals the complexity of Fermi surface topology in this system. Since then, extensive theoretical and experimental studies have been performed to provide novel insights into unconventional superconducting pairing mechanisms and physical properties in these materials. For example, the multi-gap superconductivity signals a new pathway to achieve more superconducting pairing modes, which can induce phase competition or coexistence between multiple bands by adjusting the external magnetic field or impurity distribution. Furthermore, the magnetic vortex behavior can be optimized through rational design of multi-band structures and its interaction with impurities can improve the overall performance of superconducting devices [2, 3].

As we know, each condensate in two-band superconductors is predicted to support vortex excitation with fractional quantum flux [4]. Due to the interband Josephson coupling, the vortices from different condensates are bounded together with the string interaction and their normal cores will be locked to form a composite vortex with the standard integer quantum flux in the ground state. Recently in a series of experiments of iron-based superconductors, the fractional vortices with a magnitude that varies continuously with temperature have been clearly observed in some special locations [5–7]. In general, the physics of composite vortices in the two-band system will be influenced by the coherence lengths $\xi_1$ and $\xi_2$ as well as the magnetic field penetration depth $\lambda$. When the particular condition $\xi_1 < \sqrt{2}\lambda < \xi_2$ is satisfied, there may exhibit a new superconducting state that combines characteristics of both type-1 and type-2 superconductors. This so-called semi-Meissner phase or vortex cluster phase is formed due to the interaction of long-range attraction and short-range repulsion between composite vortex excitations [8–10]. The existence of this novel vortex pattern was first visualized by Bitter decorations on high quality $MgB_2$ single crystal in 2009 [11]. Thereafter, zero-field muon spin experiments have also revealed the presence of this type-1.5 superconducting state in unconventional superconductors $Sr_2RuO_4$ [12,13] and $LaPt_3Si$ [14,15].

In the present paper, we study the electromagnetic effect of type-1.5 superconductors based on the time-dependent Ginzburg-Landau (TDGL) theory. With the COMSOL Multiphysics software and the finite element method, we first obtain the $L - \kappa_1$ phase diagram of the two-band superconductor in the absence of impurity, with $L$ the sample size and $\kappa_1$ the GL parameter. Our numerical results demonstrate that there exists a critical sample size $L_c$ for this two-band system, and the semi-Meissner state induced by long-range vortex attraction disappears below $L_c$. Then in the presence of an isotropic impurity, we show the $g - \kappa_1$ phase diagram with the sample size below $L_c$, where $g$ represents the disorder strength in this system. For $|g| > 0.22$, we can directly observe the crossover of this mesoscopic system from the diamagnetic Meissner state to the vortex cluster phase, and ultimately to the Abrikosov lattice phase. Furthermore, we also discuss the possible patterns of vortex cluster induced by the anisotropic defect structures and multiple correlated disorders in this superconductor. All of our theoretical results indicate that the diversity of impurity depositions has a significant influence on the collective behaviors of magnetic vortices in the type-1.5 superconducting

59 system.

60     The rest of this article is organized as follows. In Section 2, we introduce the two-band
61 TDGL theory and apply this formalism to the type-1.5 superconductors. In Section 3, we give
62 the procedure of numerical simulations based on the finite element method. Then in Section
63 4, we systematically investigate the impurity effect and vortex cluster phase in the mesoscopic
64 system. Finally, Section 5 gives the conclusion of the paper.

## 2   Model and formalism

66 The simplest GL free energy functional of two-gap superconductors can be written as [16–20]

$$F = \sum_i \left[ \frac{1}{2m_i} \left| \left( -i\hbar\boldsymbol{\nabla} - \frac{2e}{c}\boldsymbol{A} \right) \Psi_i \right|^2 - \alpha_i |\Psi_i|^2 + \frac{\beta_i}{2}|\Psi_i|^4 \right] + \frac{\boldsymbol{B}^2}{8\pi}. \tag{1}$$

68 Here $\Psi_i$ ($i = 1, 2$) represents the superconducting order parameter and $m_i$ is the effective
69 mass for each band. The coefficient $\alpha_i$ is a function of temperature, while $\beta_i$ is independent
70 of temperature. $\boldsymbol{B} = \boldsymbol{\nabla} \times \boldsymbol{A}$ is the magnetic induction and $\boldsymbol{A}$ is the vector potential. Starting
71 from the seminal works of Thuneberg [21,22], two main disorder models have been proposed
72 to describe the effect of nonmagnetic impurities on the superconducting system in the frame-
73 work of the GL theory [23, 24]. The first one is the $T_c$ disorder model with $T_c$ the critical
74 temperature, which is characterized by altering the GL free energy coefficient $\alpha_i \rightarrow \alpha_{i0}g(\boldsymbol{r})$
75 in Eq. (1) [25]. The other one is the $l$ disorder model with $l$ the mean free path, achieved by
76 modifying the effective mass $1/m_i \rightarrow (1/m_i)h(\boldsymbol{r})$ in Eq. (1), where $h(\boldsymbol{r}) = l/l_m < 1$ repre-
77 sents the ratio of the mean free path inside and outside the well-defined pinning area [26].

78     If the superconductor is driven out of equilibrium, the order parameter should relax back
79 to its equilibrium value. It is well known that this deviation of superconducting materials
80 can be conveniently described by the TDGL theories. The single-band TDGL equations were
81 first proposed by Schmid [27] and derived from the microscopic BCS theory by Gor'kov and
82 Éliashberg [28]. The extension of TDGL equations to the multi-component superconducting
83 system can be written as [29–32]

$$-\Gamma_i \frac{\partial \Psi_i}{\partial t} = \frac{\delta F}{\delta \Psi_i^*} \quad \text{and} \quad -\sigma_n \frac{\partial \boldsymbol{A}}{\partial t} = \frac{\delta F}{\delta \boldsymbol{A}} \tag{2}$$

84 where $\Gamma_i$ is the relaxation time of order parameters and $\sigma_n$ represents the electrical conduc-
85 tivity of the normal sample in the two-band case. Therefore, minimization of the free energy
86 $F$ with respect to $\Psi_i$ and $\boldsymbol{A}$ leads to the following dimensionless TDGL equations in the zero-
87 electrostatic potential gauge

$$-\Gamma_1 \frac{\partial \Psi_1}{\partial t} = -\left[ g(\boldsymbol{r}) - |\Psi_1|^2 \right] \Psi_1 + h(\boldsymbol{r})(-i\boldsymbol{\nabla} - \boldsymbol{A})^2 \Psi_1, \tag{3}$$

$$-\Gamma_2 \frac{\partial \Psi_2}{\partial t} = -\left[ \frac{\alpha_{20}}{\alpha_{10}} g(\boldsymbol{r}) - \frac{\beta_2}{\beta_1}|\Psi_2|^2 \right] \Psi_2 + \frac{m_1}{m_2} h(\boldsymbol{r})(-i\boldsymbol{\nabla} - \boldsymbol{A})^2 \Psi_2 \tag{4}$$

89 and

$$-\frac{\partial \boldsymbol{A}}{\partial t} = \kappa_1^2 \boldsymbol{\nabla} \times \boldsymbol{\nabla} \times \boldsymbol{A} - \boldsymbol{J}_s \tag{5}$$

90 with the supercurrent

$$\boldsymbol{J}_s = h(\boldsymbol{r}) \left\{ \left[ \frac{i}{2}(\Psi_1 \boldsymbol{\nabla} \Psi_1^* - \Psi_1^* \boldsymbol{\nabla} \Psi_1) - |\Psi_1|^2 \boldsymbol{A} \right] + \frac{m_1}{m_2} \left[ \frac{i}{2}(\Psi_2 \boldsymbol{\nabla} \Psi_2^* - \Psi_2^* \boldsymbol{\nabla} \Psi_2) - |\Psi_2|^2 \boldsymbol{A} \right] \right\}. \tag{6}$$

91 Here in the clean limit with the impurity function $g = h = 1$, we at first introduce the co-
92 herence length $\xi_i^2 = \hbar^2/(2m_i\alpha_{i0})$, the London penetration depth $\lambda^{-2} = \lambda_1^{-2} + \lambda_2^{-2}$ with
93 $\lambda_i^{-2} = 4\pi e^2 \Psi_{i0}^2/(m_i c^2)$ and $\Psi_{i0} = \sqrt{\alpha_{i0}/\beta_i}$, and the GL parameter $\kappa_1 = \lambda_1/\xi_1$. We then take
94 the coordinate $\boldsymbol{r}$ in units of $\xi_1$, the time $t$ in units of $t_0 = m_1\sigma_n/(4e^2\Psi_{10}^2)$, $\Gamma_i$ in units of
95 $\alpha_{10}t_0$ and the order parameter $\Psi_i$ in units of $\Psi_{10}$. We also set the magnetic induction $\boldsymbol{B}$ in
96 units of $H_0 = \Phi_0/(2\pi\xi_1^2)$ with the flux quantum $\Phi_0 = \pi\hbar c/e$ and the vector potential $\boldsymbol{A}$ in
97 units of $A_0 = H_0\xi_1$.

98 Following Ref. [8], multi-component systems allow a type of superconductivity that is dis-
99 tinct from type-1 or type-2 superconductor. With the condition $\xi_1 < \sqrt{2}\lambda < \xi_2$, the type-1.5
100 superconducting state will originate from a peculiar vortex interaction which exhibits short-
101 range repulsion and long-range attraction characteristics. The short-range repulsion prevents
102 adjacent vortices from overlapping, while the long-range attraction facilitates the clustering
103 of composite vortices. Consequently, this state is different from type-1 superconductors that
104 completely repel magnetic flux and type-2 superconductors which allow considerable mag-
105 netic flux penetration and the formation of vortex lattice. In the ideal sample, the constraint
106 mentioned above can be specifically expressed as

$$\sqrt{\frac{1}{2}\left(1 + \frac{m_1}{m_2}\frac{\alpha_{20}}{\alpha_{10}}\frac{\beta_1}{\beta_2}\right)} < \kappa_1 < \sqrt{\frac{1}{2}\left[\frac{m_1}{m_2}\frac{\alpha_{10}}{\alpha_{20}} + \left(\frac{m_1}{m_2}\right)^2\frac{\beta_1}{\beta_2}\right]}. \tag{7}$$

107 In this circumstance, the magnetic composite vortices will form vortex clusters and coexist
108 with domains of the two-component Meissner state in the framework of the GL theory.

109 In order to perform systematic numerical simulations, we need to specify appropriate
110 boundary conditions of the superconducting sample. We use the following superconductor-
111 insulator (or vacuum) boundary conditions in the zero-electrostatic potential gauge (see Ap-
112 pendix A for the detailed derivation) [33–35]

$$\boldsymbol{\nabla}\Psi_i \cdot \boldsymbol{n} = 0, \quad \boldsymbol{A} \cdot \boldsymbol{n} = 0 \quad \text{and} \quad \boldsymbol{\nabla} \times \boldsymbol{A} = \boldsymbol{H} \tag{8}$$

113 where $\boldsymbol{n}$ is the outward unit vector normal to the boundary and the external applied mag-
114 netic field is set as $\boldsymbol{H} = H\hat{\boldsymbol{z}}$. The first two conditions just indicate that any current passing
115 through the interface between a superconducting domain and vacuum/insulator would be
116 nonphysical for each band. The third equation represents the continuity of magnetic field
117 across the boundary. The partial differential equations (3)-(5) will be solved numerically for
118 the mesoscopic geometry in the two-dimensional space. The initial conditions at $t = 0$ are
119 taken as $|\Psi_i| = 1$ and $\boldsymbol{A} = (0, 0)$ on the $xy$-plane, corresponding to the Meissner state and
120 zero magnetic field inside the superconductor.

## 121 3  Finite element method and numerical computations

122 Based on the COMSOL Multiphysics software platform [36], we will describe the procedure
123 of the numerical simulations on the TDGL equations in this section. We first split the order
124 parameters into the real and imaginary parts, i.e. $\Psi_1 = u_1 + iu_2$ and $\Psi_2 = u_3 + iu_4$. The mag-
125 netic potential is also written in component form as $\boldsymbol{A} = (u_5, u_6)$. In order to implement the
126 boundary conditions, we will introduce an auxiliary variable $u_7(x, y, t)$ for reasons explained
127 below. In the procedure of simulations, we set $\Gamma_1 = \Gamma_2 = 5$ and $m_1 = 2m_2$. To stabilize the
128 semi-Meissner state, we also take $\alpha_{10} = \alpha_{20}$ and $\beta_1 = \beta_2$ in the calculations.

129 In this way, we can transform the TDGL equations into the general form of partial differ-

ential equations in this software package

$$\sum_k \mu_{jk}\frac{\partial u_k}{\partial t} + \sum_l \partial_l v_{jl} = \eta_j. \tag{9}$$

Here we have $j, k = 1, 2, \cdots, 7$, $l = 1, 2$ and $(\partial_1, \partial_2) = (\partial_x, \partial_y)$. The $7 \times 7$ matrix $\mu_{jk}$ and the $7 \times 2$ column vector $v_{jl}$ take the form

$$\mu_{jk} = \begin{bmatrix} 5 & 0 & 0 & 0 & 0 & 0 & 0 \\ 0 & 5 & 0 & 0 & 0 & 0 & 0 \\ 0 & 0 & 5 & 0 & 0 & 0 & 0 \\ 0 & 0 & 0 & 5 & 0 & 0 & 0 \\ 0 & 0 & 0 & 0 & 1 & 0 & 0 \\ 0 & 0 & 0 & 0 & 0 & 1 & 0 \\ 0 & 0 & 0 & 0 & 0 & 0 & 0 \end{bmatrix} \tag{10}$$

and

$$v_{jl} = \begin{bmatrix} -h(\boldsymbol{r})u_{1x} & -h(\boldsymbol{r})u_{1y} \\ -h(\boldsymbol{r})u_{2x} & -h(\boldsymbol{r})u_{2y} \\ -2h(\boldsymbol{r})u_{3x} & -2h(\boldsymbol{r})u_{3y} \\ -2h(\boldsymbol{r})u_{4x} & -2h(\boldsymbol{r})u_{4y} \\ 0 & \kappa_1^2\left(u_{6x} - u_{5y} - H\right) \\ \kappa_1^2\left(u_{5y} - u_{6x} + H\right) & 0 \\ u_5 & u_6 \end{bmatrix}. \tag{11}$$

Noting that the subscript $x$ or $y$ denotes the partial derivative with respect to the corresponding variable here. Meanwhile, the driving force $\eta_j$ contains all other terms in the TDGL equations except the left handed side of Eq. (9), and detailed calculations will give all the components explicitly as

$$\eta_1 = \left[g(\boldsymbol{r}) - \left(u_1^2 + u_2^2\right)\right]u_1 - h(\boldsymbol{r})\left[\left(u_5^2 + u_6^2\right)u_1 - \left(u_{5x} + u_{6y}\right)u_2 - 2\left(u_{2x}u_5 + u_{2y}u_6\right)\right], \tag{12}$$

$$\eta_2 = \left[g(\boldsymbol{r}) - \left(u_1^2 + u_2^2\right)\right]u_2 - h(\boldsymbol{r})\left[\left(u_5^2 + u_6^2\right)u_2 + \left(u_{5x} + u_{6y}\right)u_1 + 2\left(u_{1x}u_5 + u_{1y}u_6\right)\right], \tag{13}$$

$$\eta_3 = \left[g(\boldsymbol{r}) - \left(u_3^2 + u_4^2\right)\right]u_3 - 2h(\boldsymbol{r})\left[\left(u_5^2 + u_6^2\right)u_3 - \left(u_{5x} + u_{6y}\right)u_4 - 2\left(u_{4x}u_5 + u_{4y}u_6\right)\right], \tag{14}$$

$$\eta_4 = \left[g(\boldsymbol{r}) - \left(u_3^2 + u_4^2\right)\right]u_4 - 2h(\boldsymbol{r})\left[\left(u_5^2 + u_6^2\right)u_4 + \left(u_{5x} + u_{6y}\right)u_3 + 2\left(u_{3x}u_5 + u_{3y}u_6\right)\right], \tag{15}$$

$$\eta_5 = h(\boldsymbol{r})\left[\left(u_{2x}u_1 - u_{1x}u_2\right) + 2\left(u_{4x}u_3 - u_{3x}u_4\right) - \left(u_1^2 + u_2^2 + 2u_3^2 + 2u_4^2\right)u_5\right], \tag{16}$$

$$\eta_6 = h(\boldsymbol{r})\left[\left(u_{2y}u_1 - u_{1y}u_2\right) + 2\left(u_{4y}u_3 - u_{3y}u_4\right) - \left(u_1^2 + u_2^2 + 2u_3^2 + 2u_4^2\right)u_6\right], \tag{17}$$

$$\eta_7 = u_{5x} + u_{6y} + u_7. \tag{18}$$

Now we can examine the boundary conditions in this formalism. With the normal vector $\boldsymbol{n} = (n_1, n_2)$ and the column vector $v_{jl}$, the boundary conditions in Eq. (8) can be simply

casted into the compact form as

$$\sum_l n_l v_{jl} = 0 \qquad (19)$$

which is best suited to the COMSOL Multiphysics simulations. We also note that from the last equation ($j = 7$) in (9), our manipulations will give a trivial solution $u_7 = 0$ for this auxiliary variable and it insures the self-consistency of our problem.

COMSOL Multiphysics is a simulation platform based on the finite element method widely employed in solving coupled physical problems in engineering and fundamental research. The software numerically resolves partial differential equations by discretizing the continuous computational domain into a mesh composed of finite elements [37–39]. In two-dimensional geometries, triangular elements are generally adopted due to their adaptability to irregular boundaries and complex shapes. Based on this discretization, a local function space is constructed to approximate the field variables, typically using piecewise polynomial basis functions to maintain the continuity and stability across element interfaces [40]. To handle time-dependent problems with high accuracy and robustness, COMSOL utilizes implicit time-stepping schemes and often incorporates stable integration methods such as the backward Euler formulation. In our numerical calculations, we take the time step $\Delta t = 0.5t_0$ and treat a simulation as converged when the relative variations of the order parameter $\left|\Psi_1\right|$ between two sequential steps are smaller than $10^{-8}$. We also set the snapshot time at $t = 10^4 t_0$, which will be justified from two perspectives in Appendix B.

# 4 Results and discussions

In this section, we will set the external magnetic field $H = 0.8H_0$ and systematically explore the effects of sample boundary and impurities on various vortex excitations. Based on the TDGL theory (3)-(5), we first perform the numerical calculations to obtain the $L - \kappa_1$ and $g - \kappa_1$ phase diagrams in the two-band superconductor. Then, we investigate the effect of anisotropic defect structures and multiple correlated disorders on the patterns of magnetic vortex distributions in the mesoscopic sample.

Before the detailed numerical simulations, we would like to briefly discuss the method of determining the critical points of the phase transitions in the two-gap superconducting system. Following Ref. [41], we can identify the phase separation lines in $L - \kappa_1$ and $g - \kappa_1$ phase diagrams from the dependence of magnetization $M$ on the GL parameter $\kappa_1$ in our investigations. For the type-2 superconductor, at small $\kappa_1$ the system will stay at the Meissner phase and give the magnetization $-4\pi M = H - \langle B \rangle \approx 0.8H_0$ due to the perfect diamagnetism, where $\langle B \rangle = \langle u_{6x} - u_{5y} \rangle$ describes the average magnetic induction over the sample area $S$. While at large $\kappa_1$, the magnetic field penetrates the superconductor to form the Abrikosov vortex lattice and the magnetization will reduce gradually in a broad range of $\kappa_1$. Thus, with the definition of $M' = \mathrm{d}M/\mathrm{d}\kappa_1$, it will show a discontinuous jump at the critical $\kappa_1$ of the phase transition from the perfect diamagnetic state to the vortex lattice state. For the type-1.5 superconductor, at small $\kappa_1$ the sample remains in the Meissner state and $M$ is still close to $0.8H_0$. With the increase of $\kappa_1$, the system first enters the vortex cluster phase and the magnetization will decrease linearly in a narrow range of $\kappa_1$ exactly as in the case of the intermediate state of type-1 superconductors. As $\kappa_1$ is further raised, we will observe the vortex lattice phase, and the decline in $M$ will exhibit a significant deceleration compared to the vortex cluster phase due to the dominance of short-range repulsive intervortex interaction. In this circumstance, $M'$ will display two discontinuous jumps at the critical points of the phase transitions from the perfect diamagnetic state to the vortex cluster state, and ultimately to the vortex lattice state. We note that the identification method discussed above is still applicable

186    to systems with a relatively small number of vortices.

## 4.1    $L - \kappa_1$ phase diagram in the clean limit

188    In this subsection, we first investigate the $L - \kappa_1$ phase diagram of the $L \times L$ two-band super-
189    conductor in the absence of impurity. As an example, we choose $15\xi_1 \times 15\xi_1$ and $50\xi_1 \times 50\xi_1$
190    superconducting samples to determine the critical $\kappa_1$ in the phase transitions. We then plot the
191    variations of $M$ and its derivative with $\kappa_1$ at $t = 10^4 t_0$ in Fig. 1. From Fig. 1, we can observe
192    that the $15\xi_1 \times 15\xi_1$ system exhibits the type-2 magnetic behavior. At small $\kappa_1$, the super-
193    conductor stays at the Meissner phase and the perfect diamagnetism leads to $-4\pi M \approx 0.8H_0$.
194    While at large $\kappa_1$, the sample enters the vortex lattice phase and $M$ reduces gradually in a
195    broad range of $\kappa_1$. On the contrary, we can also see from Fig. 1 that the $50\xi_1 \times 50\xi_1$ system
196    shows the type-1.5 superconducting properties. At small $\kappa_1$, the superconductor remains at
197    the Meissner state and $M$ is still close to $0.8H_0$. With the increase of $\kappa_1$, the magnetic field
198    starts to penetrate the sample and the flux lines will exist in the form of the magnetic cluster
199    due to the long-range attractive interaction between vortices, which induces a linear decrease
200    of the magnetization in a narrow range of $\kappa_1$. As $\kappa_1$ is further raised, the number of vortices
201    in the system continues to increase and eventually forms the stable Abrikosov flux lattice. In
202    this circumstance, the rate of increase in vortex density will significantly slow down compared
203    to the vortex cluster phase due to the dominance of short-range repulsive intervortex interac-
204    tion. In order to accurately determine the critical points of the phase transitions, we further
205    calculate the first-order derivative $M'$ as a function of $\kappa_1$ in the inset of Fig. 1. It can be
206    clearly observed that for the $15\xi_1 \times 15\xi_1$ superconductor, $M'$ exhibits a discontinuous jump
207    at $\kappa_1 = 1.48$, which denotes the transition of the system from the Meissner state to the vortex
208    lattice state. For the $50\xi_1 \times 50\xi_1$ superconductor, $M'$ shows two discontinuous jumps and the
209    sample enters the vortex cluster phase at $\kappa_1 = 1.28$, then transfers to the vortex lattice phase
210    at $\kappa_1 = 1.67$.

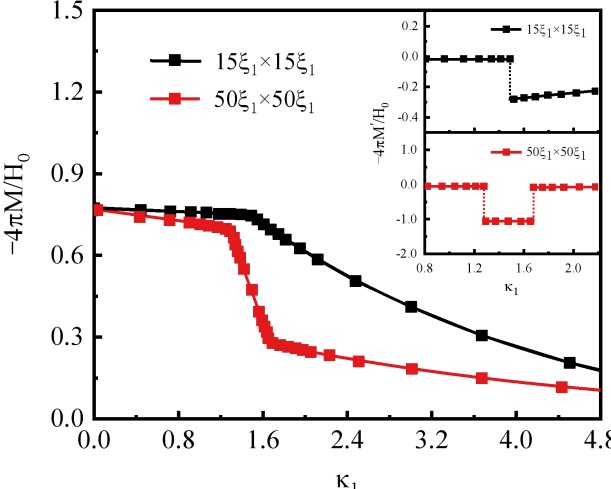

Figure 1: Variations of magnetization $M$ (main) and its first-order derivative $M'$
(inset) with GL parameter $\kappa_1$ for the $L \times L$ two-band superconductor in the absence of
impurity. We set the external magnetic field $H = 0.8H_0$ in the numerical simulations.

211    With this approach, we can further calculate the critical $\kappa_1$ for arbitrary value of $L$ and
212    obtain the $L - \kappa_1$ phase diagram as shown in Fig. 2. It can be seen from Fig. 2 that with
213    the decrease of $L$, the vortex cluster phase produced by the long-range attractive interaction

between vortices gradually vanishes. Meanwhile, we also notice the critical sample size $L_c$ for
the disappearance of this cluster state is $32\xi_1$. Thus, the superconducting system will stay in
the type-1.5 regime above $L_c$ and the type-2 regime below $L_c$ in the absence of impurity.

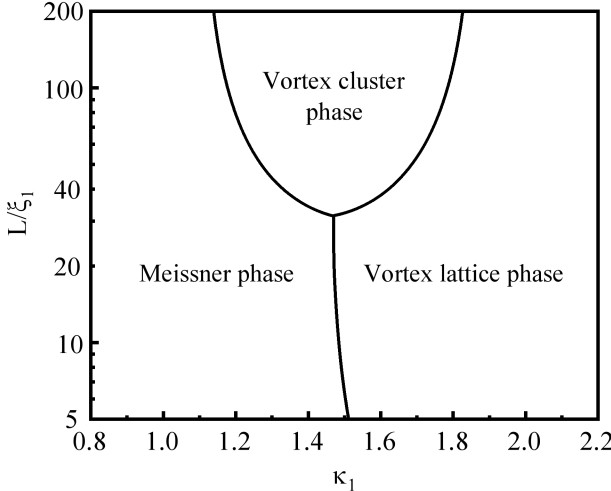

Figure 2: The $L - \kappa_1$ phase diagram of the $L \times L$ two-band superconductor in the
absence of impurity. We set the external magnetic field $H = 0.8H_0$ in the numerical
simulations, and plot the sample size $L$ on a logarithmic scale.

As we know, the type-1.5 superconductor originates from a peculiar vortex interaction that
exhibits short-range repulsion and long-range attraction characteristics. The obtained critical
$L_c$ is consistent with the characteristic length scale (about $30\xi$) of the crossover from the
attractive to repulsive intervortex interaction [9]. For the sample size $L > L_c$, the long-range
attractive potential between vortices will dominate at the external magnetic field $H = 0.8H_0$
and the system is allowed to spontaneously form the stable vortex cluster. However for $L < L_c$,
the repulsive intervortex interaction will prevail in the mesoscopic superconductor and the
vortex cluster phase can only be induced by other effects such as impurities.

In addition to the superconducting square discussed above, we further examine the transition behaviors of mesoscopic samples with the aspect ratio different from 1 in the absence of
impurity. As a simple example, we choose the $15\xi_1 \times 20\xi_1$ superconducting sample with each
side length below $L_c$. We plot the magnetic induction $B = u_{6x} - u_{5y}$ in units of $H_0$ and the
order parameter of the first condensate $\left|\Psi_1\right| = \sqrt{u_1^2 + u_2^2}$ in units of $\Psi_{10}$ at $t = 10^4 t_0$ in Fig. 3.
With the increase of the GL parameter $\kappa_1$, we can see the direct transition of this system from
the perfect diamagnetic state to the Abrikosov lattice phase as shown in Fig. 3. All of these
numerical results thus suggest that the vortex cluster phase will be excluded for arbitrary
mesoscopic sample with the characteristic scale less than $L_c$ in the absence of impurity.

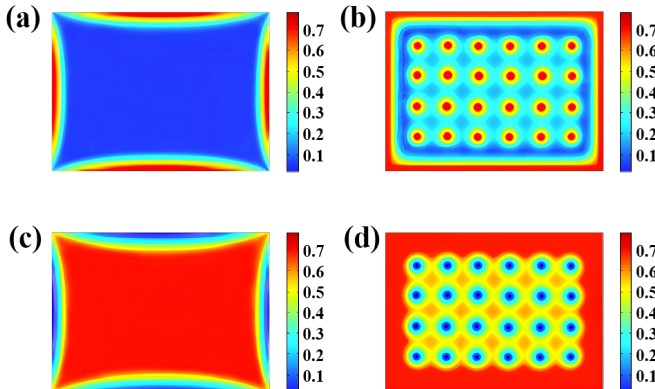

Figure 3: Transition of the magnetic induction $B$ (a,b) and the order parameter of the first condensate $\left|\Psi_1\right|$ (c,d) for the $15\xi_1 \times 20\xi_1$ type-2 superconductor. The snapshots show the Meissner phase (a,c) and vortex lattice phase (b,d) at the GL parameter $\kappa_1 = 0.70$ and $2.10$ respectively. The magnetization only has the component perpendicular to the superconducting plane.

## 4.2 Effect of an isotropic impurity in the $T_c$ and $l$ disorder models

Now, we try to explore the possible generation of the vortex cluster phase in the mesoscopic superconducting system with $L < L_c$ due to the impurity effect. As an example, we introduce an isotropic impurity with the radius $0.5\xi_1$ at the center of the $15\xi_1 \times 15\xi_1$ superconducting sample here. With the $T_c$ disorder model, the defect function $g(r)$ will be characterized by the disorder strength $g$ inside the impurity and $h(r) = 1$. We set the value of $g$ as $-0.1$ and $-0.5$, and then plot the variations of $M$ and its derivative with $\kappa_1$ at $t = 10^4 t_0$ in Fig. 4. From Fig. 4, we can observe that for $g = -0.1$, this mesoscopic system exhibits the type-2 magnetic behavior. With the transition of the superconductor from the Meissner state to the vortex lattice state, the magnetization reduces gradually from the perfect diamagnetism $-4\pi M \approx 0.8 H_0$. Meanwhile, we can also see from Fig. 4 that for $g = -0.5$, the sample shows the type-1.5 superconducting properties. In the process of the transitions from the perfect diamagnetic state to the vortex cluster state, and ultimately to the vortex lattice state, the magnetization curve first remains close to $0.8 H_0$, then decreases linearly in a narrow range of $\kappa_1$ and finally reduces with a relatively small extent compared to its neighboring phase. Furthermore, we can clearly observe from the inset of Fig. 4 that for $g = -0.1$, the phase transition from the Meissner state directly to the vortex lattice state appears at $\kappa_1 = 1.38$. For $g = -0.5$, the magnetic flux lines in this mesoscopic superconductor condense into the vortex cluster at $\kappa_1 = 1.08$ and further form the Abrikosov vortex lattice at $\kappa_1 = 1.58$.

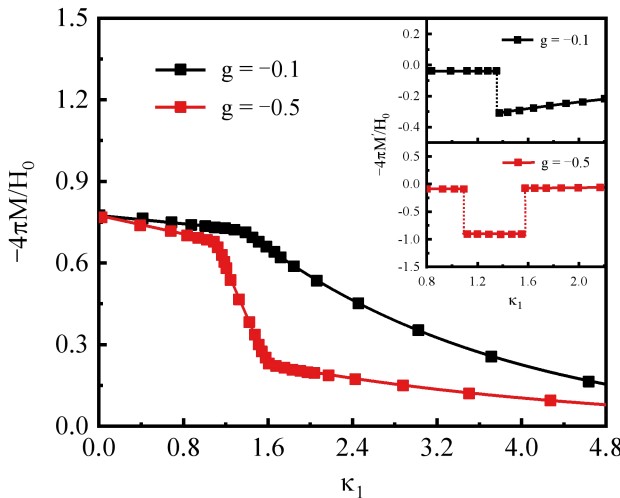

Figure 4: Variations of magnetization $M$ (main) and its first-order derivative $M'$ (inset) with GL parameter $\kappa_1$ for the $15\xi_1 \times 15\xi_1$ two-band superconductor in the presence of an isotropic impurity. We set the external magnetic field $H = 0.8H_0$ in the numerical simulations.

With this approach, we can calculate the critical $\kappa_1$ for arbitrary value of $g$ and obtain the $g - \kappa_1$ phase diagram as shown in Fig. 5. It can be seen from Fig. 5 that with the increase of the absolute value of $g$, the vortex cluster phase induced by the attractive interaction from the impurity will gradually appear in the system. Meanwhile, we also see that there exists a critical impurity strength $g_c \approx -0.22$ for the generation of the vortex cluster state in this sample. Thus, the $15\xi_1 \times 15\xi_1$ mesoscopic superconductor will stay in the type-1.5 regime for $|g| > |g_c|$ in the presence of an isotropic impurity. Furthermore, it is clearly observed that for $|g| > |g_c|$, with the increase of $|g|$ the system transfers from the Meissner phase to the vortex cluster phase at a smaller critical $\kappa_1$, and then enters the vortex lattice phase at a larger $\kappa_1$ value.

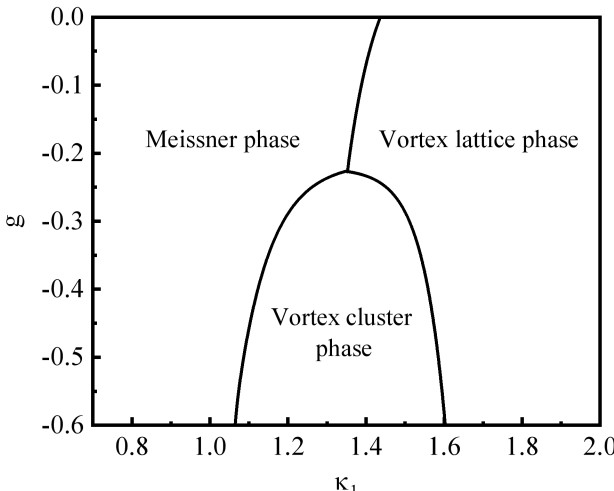

Figure 5: The $g - \kappa_1$ phase diagram of the $15\xi_1 \times 15\xi_1$ two-band superconductor in the presence of an isotropic impurity. We set the external magnetic field $H = 0.8H_0$ in the numerical simulations.

At this point, in order to demonstrate the robustness of vortex cluster phase induced by the localized impurity in the type-1.5 superconductor, we compare the numerical results computed from two types of disorder models, i.e., the $T_c$ disorder model and the $l$ disorder model. For the $T_c$ disorder model, we choose the impurity function $g$ to take the phenomenological form [25]

$$g(\mathbf{r}) = \begin{cases} -0.5, & \text{if } |\mathbf{r} - \mathbf{r}_0| < 0.5\xi_1 \\ 1, & \text{otherwise} \end{cases} \tag{20}$$

with $|g| > |g_c|$ inside the impurity. It is easy to see that this circular defect is centered at $\mathbf{r}_0 = (x_0, y_0)$. For simplicity, we insert this pinning site at the center of the $15\xi_1 \times 15\xi_1$ superconducting square. We plot the magnetic induction $B$ and the order parameter of the first condensate $|\Psi_1|$ at $t = 10^4 t_0$ in Fig. 6. With the GL parameter $\kappa_1$ taken as 0.70, 1.30 and 2.10 sequentially, we can clearly observe the transitions of this type-1.5 system from the perfect diamagnetism state to the vortex cluster phase, and ultimately to the Abrikosov lattice phase. Our numerical simulations also show that the cluster phase presents the vortex pattern with octagonal symmetry and appears in the region of $1.08 < \kappa_1 < 1.58$. Moreover, it can be seen from Fig. 6(c,f) that the isotropic defect induces the localized distortion of the Abrikosov flux lattice, but will still preserve the $C_4$ rotational symmetry of the superconducting system.

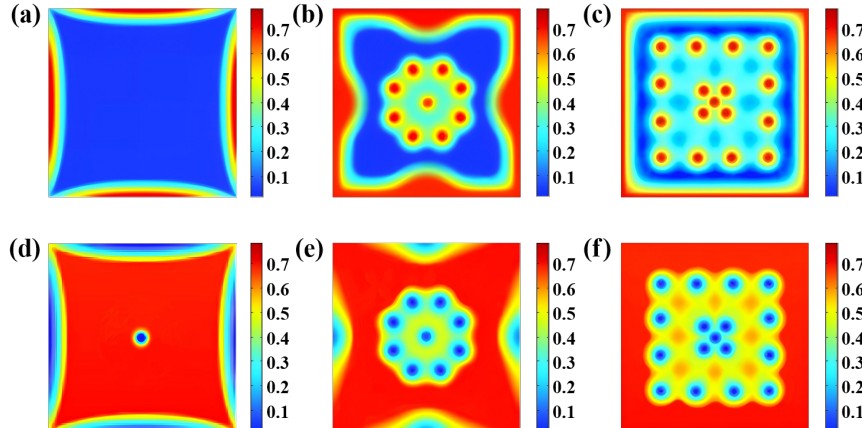

Figure 6: Transitions of the magnetic induction $B$ (a-c) and the order parameter of the first condensate $|\Psi_1|$ (d-f) for the $T_c$ disorder model at the presence of an isotropic defect in the $15\xi_1 \times 15\xi_1$ type-1.5 superconductor. The snapshots show the Meissner phase (a,d), vortex cluster phase (b,e) and vortex lattice phase (c,f) at the GL parameter $\kappa_1 = 0.70$, 1.30 and 2.10 respectively. The magnetization only has the component perpendicular to the superconducting plane.

For the $l$ disorder model, we set $g(\mathbf{r}) = 1$ and the impurity function $h$ as [26]

$$h(\mathbf{r}) = \begin{cases} 0.2, & \text{if } |\mathbf{r} - \mathbf{r}_0| < 0.5\xi_1 \\ 1, & \text{otherwise} \end{cases} \tag{21}$$

with $h < h_c$ inside the impurity. Here $h_c$ stands for the critical disorder strength for the formation of the vortex cluster state in the $l$ disorder model, which is estimated as 0.6 from our numerical simulations. Then, we also insert this pinning site at the center of the $15\xi_1 \times 15\xi_1$ mesoscopic sample. The magnetic induction $B$ and the order parameter of the first condensate $|\Psi_1|$ at $t = 10^4 t_0$ are plotted in Fig. 7. For the GL parameter $\kappa_1 = 1.30$ and 2.10, we

can observe a vortex cluster pattern with octagonal symmetry in Fig. 7(b,e) and the locally distorted flux lattice with $C_4$ rotational symmetry in Fig. 7(c,f) respectively. For this particular disorder model, the vortex cluster phase is generated around the pinning site within the range $1.06 < \kappa_1 < 1.59$. Based on the numerical results mentioned above, we can conclude that within the framework of the GL theory, the $T_c$ and $l$ disorder models are qualitatively equivalent in describing the local effect of the impurity on collective vortex distributions for the type-1.5 superconductor.

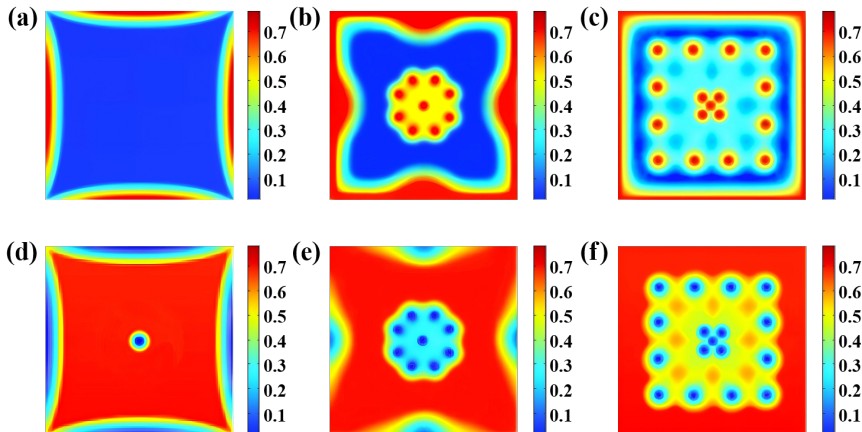

Figure 7: Transitions of the magnetic induction $B$ (a-c) and the order parameter of the first condensate $\left|\Psi_1\right|$ (d-f) for the $l$ disorder model at the presence of an isotropic defect in the $15\xi_1 \times 15\xi_1$ type-1.5 superconductor. The snapshots show the Meissner phase (a,d), vortex cluster phase (b,e) and vortex lattice phase (c,f) at the GL parameter $\kappa_1 = 0.70$, 1.30 and 2.10 respectively. The magnetization only has the component perpendicular to the superconducting plane.

At the same time, in the type-2 regime, we take the defect strength $g = -0.1$ for the $T_c$ disorder model and $h = 0.8$ for the $l$ disorder model inside each isotropic impurity. We still insert this pinning site at the center of the $15\xi_1 \times 15\xi_1$ superconducting square. We plot the magnetic induction $B$ and the order parameter of the first condensate $\left|\Psi_1\right|$ at $t = 10^4 t_0$ in Fig. 8 and Fig. 9. With the GL parameter $\kappa_1$ taken as 0.70 and 2.10 sequentially, we can observe the direct transition of this type-2 system from the perfect diamagnetic state to the Abrikosov lattice phase in Fig. 8 and Fig. 9. Based on the numerical calculations mentioned above, we can see that both the $T_c$ and $l$ disorder models give the similar results for the type-2 systems.

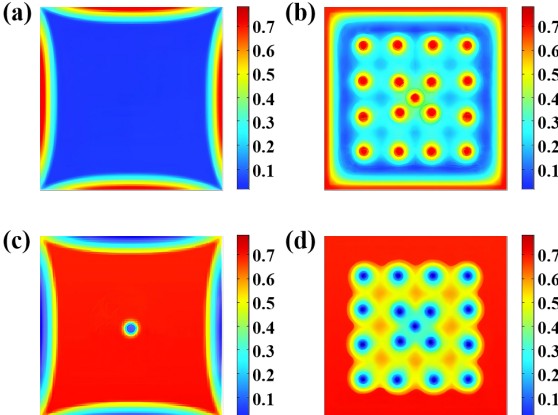

Figure 8: Transition of the magnetic induction $B$ (a,b) and the order parameter of the first condensate $\left|\Psi_1\right|$ (c,d) for the $T_c$ disorder model at the presence of an isotropic defect in the $15\xi_1 \times 15\xi_1$ type-2 superconductor. The snapshots show the Meissner phase (a,c) and vortex lattice phase (b,d) at the GL parameter $\kappa_1 = 0.70$ and 2.10 respectively. The magnetization only has the component perpendicular to the superconducting plane.

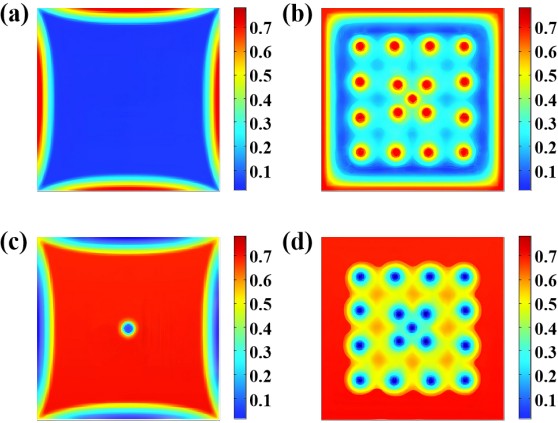

Figure 9: Transition of the magnetic induction $B$ (a,b) and the order parameter of the first condensate $\left|\Psi_1\right|$ (c,d) for the $l$ disorder model at the presence of an isotropic defect in the $15\xi_1 \times 15\xi_1$ type-2 superconductor. The snapshots show the Meissner phase (a,c) and vortex lattice phase (b,d) at the GL parameter $\kappa_1 = 0.70$ and 2.10 respectively. The magnetization only has the component perpendicular to the superconducting plane.

## 4.3 Vortex cluster phase in the presence of an anisotropic impurity

In addition to the isotropic impurity discussed above, we will investigate the effect of triangular and square defect configurations on the vortex cluster phase in the $T_c$ disorder model. The triangular or square impurity is with a side length of $\xi_1$ and placed at the center of the $15\xi_1 \times 15\xi_1$ mesoscopic superconducting system. We also set the impurity function $g = -0.5$ inside the impurity. Then, we plot the magnetic induction $B$ and the order parameter of the first condensate $\left|\Psi_1\right|$ at $t = 10^4 t_0$ for triangular and square defect configurations in Fig. 10 and Fig. 11 respectively. With the GL parameter $\kappa_1$ taken as 0.70, 1.30 and 2.10 sequentially, we can clearly observe the transitions of this system from the perfect diamagnetism state to the vortex cluster phase, and ultimately to the vortex lattice phase. For the triangular (or

square) impurity case, the peculiar vortex cluster is generated around the pinning site within the range $1.15 < \kappa_1 < 1.52$ (or $1.03 < \kappa_1 < 1.62$). It can be seen from Fig. 10(b,e) that the introduction of triangular defect breaks the $C_4$ rotational symmetry of the mesoscopic system and will form a distorted cluster in this circumstance. In contrast, the presence of square impurity ensures that the vortex pattern will still preserve the $C_4$ rotational symmetry, as shown in Figs. 11(b,e) and 11(c,f).

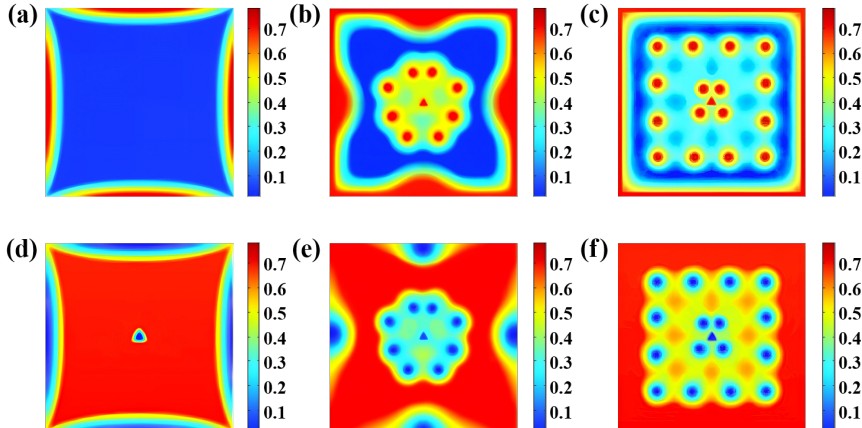

Figure 10: Transitions of the magnetic induction $B$ (a-c) and the order parameter of the first condensate $\left|\Psi_1\right|$ (d-f) at the presence of a triangular defect in the $15\xi_1 \times 15\xi_1$ type-1.5 superconductor. The snapshots show the Meissner phase (a,d), vortex cluster phase (b,e) and vortex lattice phase (c,f) at the GL parameter $\kappa_1 = 0.70$, $1.30$ and $2.10$ respectively. The magnetization only has the component perpendicular to the superconducting plane.

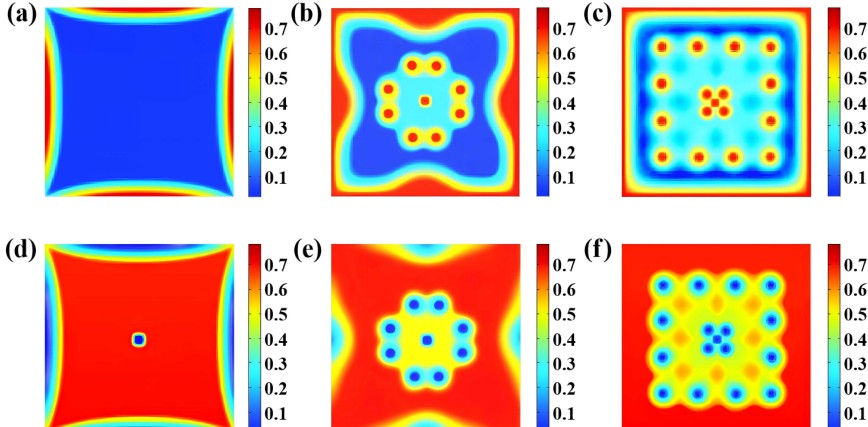

Figure 11: Transitions of the magnetic induction $B$ (a-c) and the order parameter of the first condensate $\left|\Psi_1\right|$ (d-f) at the presence of a square defect in the $15\xi_1 \times 15\xi_1$ type-1.5 superconductor. The snapshots show the Meissner phase (a,d), vortex cluster phase (b,e) and vortex lattice phase (c,f) at the GL parameter $\kappa_1 = 0.70$, $1.30$ and $2.10$ respectively. The magnetization only has the component perpendicular to the superconducting plane.

### 4.4 Uncorrelated and correlated disorder systems

In this subsection, we set the disorder strength $|g| > |g_c|$ at the impurity sites in the $T_c$ disorder model, and discuss the effects of multiple uncorrelated and correlated defects on vortex cluster patterns in the $15\xi_1 \times 15\xi_1$ mesoscopic superconductor. In the uncorrelated case, we choose the impurity function $g(\boldsymbol{r})$ to take the phenomenological form

$$g(\boldsymbol{r}) = \prod_{n=1}^{N} g_n(\boldsymbol{r}) \quad \text{with} \quad g_n(\boldsymbol{r}) = \begin{cases} -0.5, & \text{if } |\boldsymbol{r} - \boldsymbol{r}_{0n}| < 0.5\xi_1 \\ \\ 1, & \text{otherwise} \end{cases} . \tag{22}$$

It is easy to see that the isotropic impurity is centered at $\boldsymbol{r}_{0n} = (x_{0n}, y_{0n})$ with $n = 1, 2, ..., N$. For simplicity, we take the impurity number $N = 2$ and select the pinning centers at $(\pm 3\xi_1, 0)$ in $15\xi_1 \times 15\xi_1$ superconducting sample, which ensures the uncorrelation between these two defects. We plot the magnetic induction $B$ and the order parameter of the first condensate $|\Psi_1|$ at $t = 10^4 t_0$ in Fig. 12. Different from the single impurity case, multiple vortex clusters are generated around the pinning sites within $0.87 < \kappa_1 < 1.77$. With the GL parameter $\kappa_1 = 1.30$, we can see from Fig. 12(b,e) that each vortex cluster exhibits the identical pattern with hexagonal symmetry. Meanwhile for $\kappa_1 = 2.10$, as shown in Fig. 12(c,f), we can clearly observe the localized distortions around the pinning positions in the flux lattice due to the attraction of vortices by impurities.

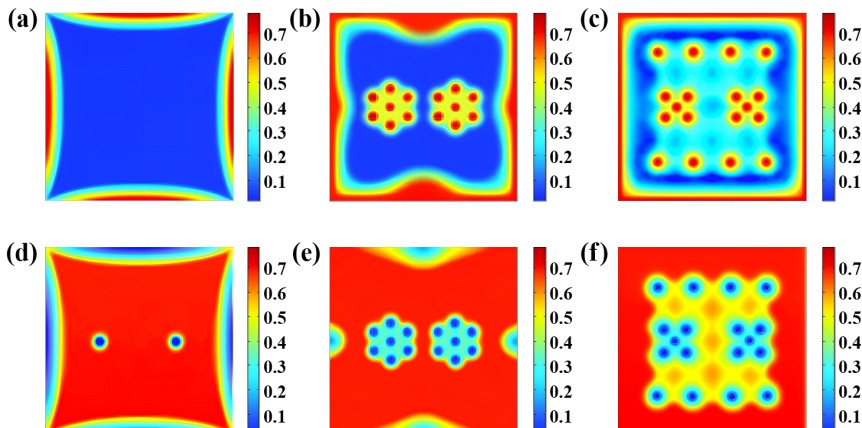

Figure 12: Transitions of the magnetic induction $B$ (a-c) and the order parameter of the first condensate $|\Psi_1|$ (d-f) at the presence of two uncorrelated defects in the $15\xi_1 \times 15\xi_1$ type-1.5 superconductor. The snapshots show the Meissner phase (a,d), vortex cluster phase (b,e) and vortex lattice phase (c,f) at the GL parameter $\kappa_1 = 0.70$, 1.30 and 2.10 respectively. The magnetization only has the component perpendicular to the superconducting plane.

In order to take into account the spatial correlation between these impurities, we choose the following continuous pinning function [42, 43]

$$g(\boldsymbol{r}) = \prod_{n=1}^{N} g_n(\boldsymbol{r}) \quad \text{with} \quad g_n(\boldsymbol{r}) = \tanh\left(\frac{|\boldsymbol{r} - \boldsymbol{r}_{0n}| - R}{R_0}\right). \tag{23}$$

We take $R = 0.5\xi_1$ and $R_0 = 1.5\xi_1$ in Eq. (23), and then perform numerical simulations in the $15\xi_1 \times 15\xi_1$ mesoscopic superconductor. For comparison with the uncorrelated case, we still choose the defect centers at $(\pm 3\xi_1, 0)$. We plot the magnetic induction $B$ and the

order parameter of the first condensate $\left|\Psi_1\right|$ at $t = 10^4 t_0$ in Fig. 13. Note that with this new impurity function, we can obtain the stable vortex cluster phase within $0.92 < \kappa_1 < 1.71$. For $\kappa_1 = 1.30$, it is shown in Fig. 13(b,e) that two vortex clusters induced by uncorrelated disorders in Fig. 12(b,e) are fused into a single larger cluster here. Meanwhile with $\kappa_1 = 2.10$, we can find a vortex lattice pattern with local distortions around the impurities in Fig. 13(c,f).

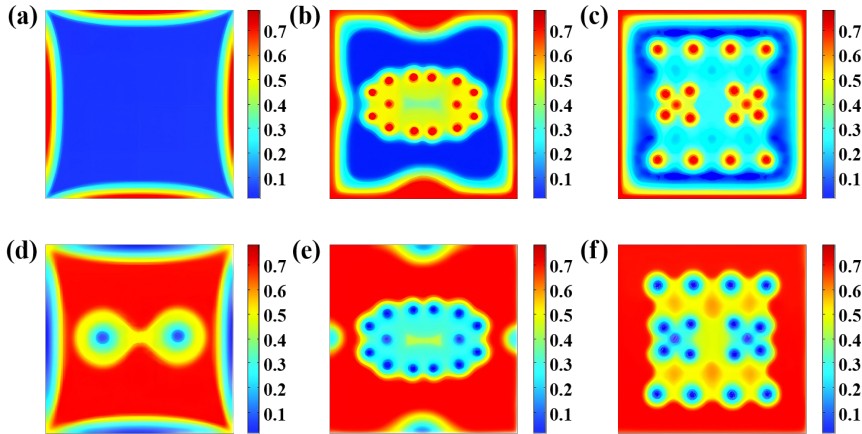

Figure 13: Transitions of the magnetic induction $B$ (a-c) and the order parameter of the first condensate $\left|\Psi_1\right|$ (d-f) at the presence of two correlated defects in the $15\xi_1 \times 15\xi_1$ type-1.5 superconductor. The snapshots show the Meissner phase (a,d), vortex cluster phase (b,e) and vortex lattice phase (c,f) at the GL parameter $\kappa_1 = 0.70$, 1.30 and 2.10 respectively. The magnetization only has the component perpendicular to the superconducting plane.

## 5    Conclusion

Based on two-band TDGL theory, we explore the impurity effect on the vortex collective behaviors in the mesoscopic type-1.5 superconductor. With the finite element method, the investigations suggest that the vortex cluster phase will be excluded for arbitrary mesoscopic sample with the characteristic scale less than $L_c$ in the absence of impurity. In the presence of an isotropic impurity, our numerical results give the direct evidence for the existence of semi-Meissner state at $|g| > |g_c|$ due to the attractive defect interaction. We also discuss the effect of anisotropic defect structures and multiple correlated disorders on the possible patterns of magnetic vortex distributions. We hope that our theoretical results will inspire further research on better understanding novel vortex dynamics and transport properties in two-band superconductors.

## Acknowledgements

One of the authors (H.H.) would like to thank Prof. Z.-Z. Gan for helpful discussions. We would also like to thank the anonymous referees for very inspiring suggestions.

**Author contributions**    Conceptualization, T.H., G.W., J.L. and H.H.; methodology, T.H., G.W., J.L. and H.H.; software, T.H., G.W.; validation, G.W., J.L. and H.H.; formal analysis, T.H., G.W., J.L. and H.H.; investigation, T.H., G.W., J.L. and H.H.; resources, H.H.; data curation, T.H., G.W., J.L. and H.H.; writing-original draft preparation, T.H., G.W. and H.H.; writing-review

358 and editing, T.H., G.W., J.L. and H.H.; visualization, T.H.; supervision, G.W., J.L. and H.H.;
359 project administration, H.H. All authors have read and agreed to the published version of the
360 manuscript.

# A  Zero electric potential gauge and boundary conditions

362 In this Appendix, we will discuss a particular gauge choice of the boundary conditions and
363 present a microscopic derivation of the dimensionless boundary condition $(\nabla - \mathrm{i}A)\Psi_i \cdot n = 0$.
364 Firstly, we try to show that in the zero electric potential gauge, the dimensionless boundary
365 conditions will take the form

$$\nabla\Psi_i \cdot n = 0, \quad A \cdot n = 0 \quad \text{and} \quad \nabla \times A = H \tag{A.1}$$

366 as adopted in Eq. (8).
367    We start from the following gauge invariant boundary conditions between a two-band
368 superconductor and an insulator (or vacuum)

$$(\nabla - \mathrm{i}A)\Psi_i \cdot n = 0, \quad \left(\frac{\partial A}{\partial t} + \nabla\varphi\right) \cdot n = 0 \quad \text{and} \quad \nabla \times A = H. \tag{A.2}$$

369 Here $\varphi$ is defined as the electric potential. Given an arbitrary function $\chi$, the gauge transfor-
370 mation takes the form as

$$\Psi_i \to \Psi_i e^{\mathrm{i}\chi}, \quad A \to A + \nabla\chi \quad \text{and} \quad \varphi \to \varphi - \frac{\partial\chi}{\partial t}. \tag{A.3}$$

371 It is easy to show that the boundary conditions in Eq. (A.2) maintain the gauge invariance.
372 Then with the zero electric potential gauge, we can get from the transformation in Eq. (A.3)

$$\frac{\partial\chi}{\partial t} = \varphi. \tag{A.4}$$

373 Plugging this condition into the second equation of the boundary conditions (A.2), it leads to

$$\frac{\partial A}{\partial t} \cdot n = 0 \tag{A.5}$$

374 in this new gauge. This equation can be integrated to give $A \cdot n = 0$, which transforms the
375 boundary condition $(\nabla - \mathrm{i}A)\Psi_i \cdot n = 0$ into the form $\nabla\Psi_i \cdot n = 0$. Based on the analysis above,
376 we can see that the boundary conditions in Eq. (A.1) are simply the result of a particular gauge
377 choice.
378    Secondly, we would like to give a microscopic derivation of the dimensionless boundary
379 condition $(\nabla - \mathrm{i}A)\Psi_i \cdot n = 0$ which is presented in Eq. (A.2). We try to show that at the
380 interface of two-band superconductor and insulator (or vacuum), this boundary condition
381 is applicable not only for the simple U(1)×U(1) symmetric model studied here but also the
382 two-component GL models in general. In this process, we will follow the procedure in the
383 single-band case suggested by de Gennes [44].
384    Based on the work of Zhitomirsky and Dao [45], we write the Hamiltonian of a two-band
385 superconductor as

$$H = \sum_{i\sigma} c_{i\sigma}^\dagger(r)\hat{h}(r)c_{i\sigma}(r) - \sum_{ii'} g_{ii'} c_{i\uparrow}^\dagger(r)c_{i\downarrow}^\dagger(r)c_{i'\downarrow}(r)c_{i'\uparrow}(r). \tag{A.6}$$

386 Here, $i, i' = 1, 2$ are the band indices and $\sigma = \uparrow, \downarrow$ is the spin index. $\hat{h}(r)$ is the single par-
387 ticle Hamiltonian of the normal metal, and $g_{ii'}$ are the effective electron-electron interaction
388 constants with $g_{12} = g_{21}$.

389    We can introduce the gap functions

$$\Delta_i(\boldsymbol{r}) = -\sum_{i'} g_{ii'} \left\langle c_{i'\downarrow}(\boldsymbol{r})c_{i'\uparrow}(\boldsymbol{r}) \right\rangle \tag{A.7}$$

390    and transform the Hamiltonian into the mean field form

$$H_{\text{eff}} = \sum_{i\sigma} c_{i\sigma}^\dagger(\boldsymbol{r})\hat{h}(\boldsymbol{r})c_{i\sigma}(\boldsymbol{r}) + \sum_i \left[ \Delta_i(\boldsymbol{r})c_{i\uparrow}^\dagger(\boldsymbol{r})c_{i\downarrow}^\dagger(\boldsymbol{r}) + \text{H.c.} \right]. \tag{A.8}$$

391    This effective Hamiltonian can be diagonalized by means of the Bogoliubov transformation
392    with $b$ and $b^\dagger$ the annihilation and creation operators of quasi-particle excitations

$$c_{i\uparrow}(\boldsymbol{r}) = \sum_{\boldsymbol{k}} \left[ u_{i\boldsymbol{k}}(\boldsymbol{r})b_{i\boldsymbol{k}\uparrow} - v_{i\boldsymbol{k}}^*(\boldsymbol{r})b_{i\boldsymbol{k}\downarrow}^\dagger \right] \tag{A.9}$$

393    and

$$c_{i\downarrow}(\boldsymbol{r}) = \sum_{\boldsymbol{k}} \left[ u_{i\boldsymbol{k}}(\boldsymbol{r})b_{i\boldsymbol{k}\downarrow} + v_{i\boldsymbol{k}}^*(\boldsymbol{r})b_{i\boldsymbol{k}\uparrow}^\dagger \right] \tag{A.10}$$

394    where $\boldsymbol{k}$ is the wave vector. With the anti-commutation relations between the fermion opera-
395    tors and the equation of motion for $c_{i\sigma}(\boldsymbol{r})$, we can obtain the Bogoliubov-de Gennes equations
396    for a two-band superconductor

$$\begin{pmatrix} \hat{h} & \Delta_i(\boldsymbol{r}) \\ \Delta_i^*(\boldsymbol{r}) & -\hat{h}^* \end{pmatrix} \begin{pmatrix} u_{i\boldsymbol{k}}(\boldsymbol{r}) \\ v_{i\boldsymbol{k}}(\boldsymbol{r}) \end{pmatrix} = E_{i\boldsymbol{k}} \begin{pmatrix} u_{i\boldsymbol{k}}(\boldsymbol{r}) \\ v_{i\boldsymbol{k}}(\boldsymbol{r}) \end{pmatrix} \tag{A.11}$$

397    where $E_{i\boldsymbol{k}}$ is the energy of the excitation. Then with Eq. (A.7), we can transform the self-
398    consistent gap equations into

$$\Delta_i(\boldsymbol{r}) = \sum_{i'\boldsymbol{k}} g_{ii'} v_{i'\boldsymbol{k}}^*(\boldsymbol{r}) u_{i'\boldsymbol{k}}(\boldsymbol{r}) \left[ 1 - 2f(E_{i'\boldsymbol{k}}) \right] \tag{A.12}$$

399    with $f(E_{i\boldsymbol{k}}) = \left[ 1 + \exp(E_{i\boldsymbol{k}}/k_B T) \right]^{-1}$ and $T$ the temperature.
400    In the analogy with the single-band case, for small gap functions $\Delta_i$, we can obtain the
401    linearized form of self-consistency conditions from Eqs. (A.11) and (A.12) as

$$\Delta_i(\boldsymbol{r}) = \sum_{i'} \int K_{ii'}(\boldsymbol{r}, \boldsymbol{r}')\Delta_{i'}(\boldsymbol{r}')\mathrm{d}\boldsymbol{r}' \tag{A.13}$$

402    with the kernel

$$K_{ii'}(\boldsymbol{r}, \boldsymbol{r}') = \frac{g_{ii'}}{2} \sum_{\boldsymbol{k}\boldsymbol{k}'} \frac{\tanh\left(\frac{\varepsilon_{i'\boldsymbol{k}}}{2k_B T}\right) + \tanh\left(\frac{\varepsilon_{i'\boldsymbol{k}'}}{2k_B T}\right)}{\varepsilon_{i'\boldsymbol{k}} + \varepsilon_{i'\boldsymbol{k}'}} \Phi_{i'\boldsymbol{k}}^*(\boldsymbol{r}') \Phi_{i'\boldsymbol{k}'}^*(\boldsymbol{r}') \Phi_{i'\boldsymbol{k}}(\boldsymbol{r}) \Phi_{i'\boldsymbol{k}'}(\boldsymbol{r}). \tag{A.14}$$

403    Here $\Phi_{i'\boldsymbol{k}}(\boldsymbol{r})$ and $\varepsilon_{i'\boldsymbol{k}}$ are defined as the normal-state eigenfunction and eigenvalue of the
404    electron with $\hat{h}\Phi_{i'\boldsymbol{k}} = \varepsilon_{i'\boldsymbol{k}}\Phi_{i'\boldsymbol{k}}$.
405    We now assume the small spatial variations in the vector potential $\boldsymbol{A}$. Then the eigenfunc-
406    tions $\Phi_{i'\boldsymbol{k}}$ in the normal metal in the presence of $\boldsymbol{A}$ will differ from the eigenfunctions $w_{i'\boldsymbol{k}}$ in
407    the absence of $\boldsymbol{A}$ by only a phase factor, i.e.,

$$\Phi_{i'\boldsymbol{k}}^*(\boldsymbol{r}')\Phi_{i'\boldsymbol{k}}(\boldsymbol{r}) \rightarrow w_{i'\boldsymbol{k}}^*(\boldsymbol{r}')w_{i'\boldsymbol{k}}(\boldsymbol{r}) \exp\left[ \frac{\mathrm{i}}{2}\boldsymbol{A} \cdot (\boldsymbol{r} - \boldsymbol{r}') \right]. \tag{A.15}$$

408    Plugging into Eq. (A.13), it will lead to

$$\Delta_i(\boldsymbol{r}) = \sum_{i'} \int \overline{K}_{ii'}(\boldsymbol{r}, \boldsymbol{r}')\Delta_{i'}(\boldsymbol{r}') \exp\left[ \mathrm{i}\boldsymbol{A} \cdot (\boldsymbol{r} - \boldsymbol{r}') \right] \mathrm{d}\boldsymbol{r}' \tag{A.16}$$

with the kernel in the absence of external magnetic field

$$\overline{K}_{ii'}(\boldsymbol{r},\boldsymbol{r}') = \frac{g_{ii'}}{2} \sum_{\boldsymbol{k}\boldsymbol{k}'} \frac{\tanh\left(\frac{\varepsilon_{i'\boldsymbol{k}}}{2k_B T}\right) + \tanh\left(\frac{\varepsilon_{i'\boldsymbol{k}'}}{2k_B T}\right)}{\varepsilon_{i'\boldsymbol{k}} + \varepsilon_{i'\boldsymbol{k}'}} w_{i'\boldsymbol{k}}^*(\boldsymbol{r}')\, w_{i'\boldsymbol{k}'}^*(\boldsymbol{r}')\, w_{i'\boldsymbol{k}}(\boldsymbol{r})\, w_{i'\boldsymbol{k}'}(\boldsymbol{r}). \tag{A.17}$$

Thus from Eq. (A.16), we can write

$$\Delta_i(\boldsymbol{r}) = \overline{\Delta}_i(\boldsymbol{r}) \exp(\mathrm{i}\boldsymbol{A}\cdot\boldsymbol{r}) \tag{A.18}$$

with $\overline{\Delta}_i(\boldsymbol{r})$ the superconducting gap function in the absence of $\boldsymbol{A}$. Then we have

$$\overline{\Delta}_i(\boldsymbol{r}) = \sum_{i'} \int \overline{K}_{ii'}(\boldsymbol{r},\boldsymbol{r}')\overline{\Delta}_{i'}(\boldsymbol{r}')\mathrm{d}\boldsymbol{r}'. \tag{A.19}$$

Now, we can examine the behavior of the superconducting gap functions near the superconductor-insulator interface. Following the procedure pioneered by de Gennes, we suppose that the gap functions close to the surface behaves as

$$\overline{\Delta}_i(s) = \overline{\Delta}_{i0} + \left(\sum_{i'} \frac{\xi_1}{b_{ii'}}\overline{\Delta}_{i'0}\right) s. \tag{A.20}$$

Here $s$ measures the normal distance from the boundary in units of $\xi_1$ and $s > 0$ is defined in the superconductor. For simplicity, we set the cross section of the boundary as 1. $\overline{\Delta}_{i0}$ represents the gap function at the boundary and $b_{ii'}$ denotes the intraband or interband surface extrapolation length for the two-band superconductor. From Eq. (A.20), we can establish the boundary condition between the two-band superconductor and the insulator (or vacuum) at $s = 0$

$$\frac{\mathrm{d}\overline{\Delta}_i}{\mathrm{d}s} = \sum_{i'} \frac{\xi_1}{b_{ii'}}\overline{\Delta}_{i'} \tag{A.21}$$

in the absence of external magnetic field.

Meanwhile, with the explicit expressions of the kernels in the bulk system and the addition of nonlinear terms to the gap equations, we can obtain the two-band GL equations from Eq. (A.19) as [45]

$$-\alpha_1\overline{\Delta}_1 + \beta_1\left|\overline{\Delta}_1\right|^2\overline{\Delta}_1 - \gamma_1\boldsymbol{\nabla}^2\overline{\Delta}_1 - R_{12}\overline{\Delta}_2 = 0 \tag{A.22}$$

and

$$-\alpha_2\overline{\Delta}_2 + \beta_2\left|\overline{\Delta}_2\right|^2\overline{\Delta}_2 - \gamma_2\boldsymbol{\nabla}^2\overline{\Delta}_2 - R_{12}\overline{\Delta}_1 = 0, \tag{A.23}$$

with the GL parameters

$$\alpha_{1,2} = N_{1,2}\left[\frac{1}{\lambda_{\max}} - \frac{\lambda_{22,11}}{\lambda} + \ln\left(\frac{T_{c0}}{T}\right)\right], \quad \beta_i = \frac{7\zeta(3)N_i}{16\pi^2(k_B T_{c0})^2}, \tag{A.24}$$

$$\gamma_i = \frac{7\zeta(3)\hbar^2 N_i v_{Fi}^2}{16\pi^2(k_B T_{c0})^2} \quad \text{and} \quad R_{12} = \frac{N_1\lambda_{12}}{\lambda} = \frac{N_2\lambda_{21}}{\lambda}.$$

Here $\lambda_{ii'} = g_{ii'}N_{i'}$ with $N_{i'}$ the density of states at the Fermi level for each band, $\lambda = \lambda_{11}\lambda_{22} - \lambda_{12}\lambda_{21}$ and $\lambda_{\max} = \frac{1}{2}\left[(\lambda_{11}+\lambda_{22}) + \sqrt{(\lambda_{11}-\lambda_{22})^2 + 4\lambda_{12}\lambda_{21}}\right]$ the largest eigenvalue of $\lambda$-matrix. $T_{c0}$ is the bulk critical temperature and $v_{Fi}$ is the average Fermi velocity for each band.

431 In the spatially homogeneous case, we can neglect the gradient $\gamma$-terms. Eqs. (A.22) and
432 (A.23) yield the gap equation at $T = T_{c0}$

$$\begin{pmatrix} \lambda_{11} & \lambda_{12} \\ \lambda_{21} & \lambda_{22} \end{pmatrix} \begin{pmatrix} \overline{\Delta}_1 \\ \overline{\Delta}_2 \end{pmatrix} = \lambda_{\max} \begin{pmatrix} \overline{\Delta}_1 \\ \overline{\Delta}_2 \end{pmatrix}, \tag{A.25}$$

433 which obviously gives the consistent result.

434 Now, we try to determine the coefficients $b_{ii'}$ in Eq. (A.21) by solving the linearized gap
435 equation (A.19) in absence of external magnetic field. If we introduce $\overline{K}^0_{ii'}(s,s')$ as the kernel
436 of gap functions in the superconducting bulk system, we can transform Eq. (A.19) into

$$\overline{\Delta}_i(s) - \sum_{i'} \int \overline{K}^0_{ii'}(s,s')\overline{\Delta}_{i'}(s')\mathrm{d}s' = -\sum_{i'} \int \left[ \overline{K}^0_{ii'}(s,s') - \overline{K}_{ii'}(s,s') \right] \overline{\Delta}_{i'}(s')\mathrm{d}s' \equiv -\sum_{i'} H_{ii'}(s). \tag{A.26}$$

437

438 From Eqs. (A.22) and (A.23) with the higher order $\beta$-terms omitted, also noting that
439 $\overline{K}^0_{ii'}(s,s') = \overline{K}^0_{ii'}(s-s')$ due to the translational symmetry, we can read out the Laplace trans-
440 formation of $\overline{K}^0_{ii'}$ as

$$\overline{K}^0_{ii'}(p) = \frac{\lambda_{ii'}}{\lambda_{\max}} + \frac{\lambda_{ii'}\gamma_{i'}}{N_{i'}\xi_1^2}p^2. \tag{A.27}$$

441 Plugging Eq. (A.27) into (A.26), we can get

$$\overline{\Delta}_i(p) - \sum_{i'} \left( \frac{\lambda_{ii'}}{\lambda_{\max}} \right) \overline{\Delta}_{i'}(p) - \sum_{i'} \left( \frac{\lambda_{ii'}\gamma_{i'}}{N_{i'}\xi_1^2} \right) p^2\overline{\Delta}_{i'}(p) = -\sum_{i'} H_{ii'}(p). \tag{A.28}$$

442 Here $\overline{\Delta}_i(p)$ and $H_{ii'}(p)$ are the Laplace transformations of $\overline{\Delta}_i(s)$ and $H_{ii'}(s)$ respectively. Since
443 the first two terms of the left-handed side in Eq. (A.28) can be cancelled out according to Eq.
444 (A.25), we then have

$$\sum_{i'} \left( \frac{\lambda_{ii'}\gamma_{i'}}{N_{i'}\xi_1^2} \right) p^2\overline{\Delta}_{i'}(p) = \sum_{i'} H_{ii'}(p). \tag{A.29}$$

445 We can see that both sides in Eq. (A.29) take the main contribution from the boundary region.
446 Notice that the Laplace transformation of the gap function in Eq. (A.20) takes the form

$$\overline{\Delta}_i(p) = \frac{\overline{\Delta}_{i0}}{p} + \sum_{i'} \frac{\xi_1\overline{\Delta}_{i'0}}{b_{ii'}p^2}. \tag{A.30}$$

447 Then at $p \to 0$, we will obtain from Eq. (A.29)

$$\sum_{i'i''} \left( \frac{\lambda_{ii'}\gamma_{i'}}{N_{i'}\xi_1 b_{i'i''}} \right) \overline{\Delta}_{i''0} = \sum_{i'} H_{ii'}(p=0). \tag{A.31}$$

448 Parallel to de Gennes' analysis, we have the sum rules

$$\int \overline{K}^0_{ii'}(s,s')\mathrm{d}s' = \frac{\lambda_{ii'}}{\lambda_{\max}} \quad \text{and} \quad \int \overline{K}_{ii'}(s,s')\mathrm{d}s' = \frac{\lambda_{ii'}N_{i'}(s)}{\lambda_{\max}N_{i'}} \tag{A.32}$$

449 with $N_{i'}(s)$ the local density of states at the Fermi surface. Then, we can write the Laplace
450 transformation of the kernel difference at $p \to 0$

$$H_{ii'}(p=0) = \int H_{ii'}(s)\mathrm{d}s = \frac{\lambda_{ii'}\overline{\Delta}_{i'0}}{\lambda_{\max}} \int \frac{\overline{\Delta}_{i'}(s)}{\overline{\Delta}_{i'0}} \left[ 1 - \frac{N_{i'}(s)}{N_{i'}} \right] \mathrm{d}s. \tag{A.33}$$

Now we suppose $\overline{\Delta}_{i'}(s)/\overline{\Delta}_{i'0}$ approaches zero in the insulating region and is of the order of 1 in the metallic region. $N_{i'}(s)/N_{i'}$ also passes from $0 \rightarrow 1$ in a few interatomic distances from the boundary. Therefore, the integrand in Eq. (A.33) is nonvanishing only in a width of order of the lattice constant $a$. We can then estimate $H_{ii'}(p=0)$ as

$$H_{ii'}(p=0) = \frac{\lambda_{ii'}a}{\lambda_{\max}\xi_1}\overline{\Delta}_{i'0}. \qquad (A.34)$$

Comparing Eq. (A.31) with Eq. (A.34), we can finally obtain

$$\frac{1}{b_{ii}} = \frac{N_i a}{\gamma_i \lambda_{\max}} \quad \text{and} \quad \frac{1}{b_{12}} = \frac{1}{b_{21}} = 0. \qquad (A.35)$$

At this stage, we would like to point out that $1/b_{ii'} = 0$ ($i \neq i'$) is only an approximation and will become nonzero in the higher-order calculation. Even for a contact between a superconductor and an insulator, the Cooper pairs can still diffuse into the insulating region with some probability. Algebraically, this means that the gap function $\overline{\Delta}_{i'}(s)$ will also extend into the $s < 0$ region, and we can roughly estimate $\overline{\Delta}_{i'}(s) \sim \sum_{i''} T_{i'i''}\overline{\Delta}_{i''0}e^{\xi_1 s/a}$ ($s < 0$) with $T_{i'i''}$ the element of the transmission matrix at the boundary. Including the $s < 0$ part in the integration of Eq. (A.33) and noting $N_{i'}(s)/N_{i'} \approx 0$ in this region, we can get $H_{ii'}(p=0) = (\lambda_{ii'}a/\lambda_{\max}\xi_1)(\overline{\Delta}_{i'0} + \sum_{i''} T_{i'i''}\overline{\Delta}_{i''0})$. Plugging into Eq. (A.31), the coefficients of boundary terms are given by

$$\frac{1}{b_{ii}} = \frac{N_i a}{\gamma_i \lambda_{\max}}\left(1 + T_{ii}\right), \quad \frac{1}{b_{12}} = \frac{N_1 a}{\gamma_1 \lambda_{\max}}T_{12} \quad \text{and} \quad \frac{1}{b_{21}} = \frac{N_2 a}{\gamma_2 \lambda_{\max}}T_{21}. \qquad (A.36)$$

With the transmission coefficient from the superconductor to the insulator $T_{ii'} \ll 1$, we can obviously see that Eq. (A.35) is a good approximation.

For a typical two-band superconductor, we can estimate $\gamma_i \lambda_{\max}/N_i \sim \xi_1^2$ with $\xi_1 \sim 10^{-4}$ cm and the lattice constant $a \sim 10^{-8}$ cm, which will give $b_{ii} \sim 1$ cm. Therefore for a boundary separating a two-band superconductor from an insulator we can set $\xi_1/b_{ii'} \approx 0$. This leads to the boundary condition $d\overline{\Delta}_i/ds = 0$ from Eq. (A.21). For an arbitrary superconducting domain and in the presence of the magnetic field, we can generalize this result to $(\boldsymbol{\nabla} - i\mathbf{A})\Delta_i \cdot \mathbf{n} = 0$ according to Eq. (A.18). With the phenomenological superconducting order parameter $\Psi_i \propto \Delta_i$, we can finally write down the boundary condition $(\boldsymbol{\nabla} - i\mathbf{A})\Psi_i \cdot \mathbf{n} = 0$ for the interface of two-band superconductor and insulator.

# B  Discussion on convergence and relaxation time in numerical simulations

In this Appendix, we would like to justify the choice of the snapshot time at $t = 10^4 t_0$ in our numerical simulations from two perspectives. On one hand, we take the time step $\Delta t = 0.5 t_0$ in our numerical calculations and treat a simulation as converged when the relative variation of the order parameter $\left|\Psi_1\right|$ between two sequential steps is smaller than $10^{-8}$. Our computational results indicate that for the $15\xi_1 \times 15\xi_1$ superconducting systems with different defect configurations, the system will consistently reach the convergence before the snapshot time $10^4 t_0$. On the other hand, we can define an average velocity $\overline{v} = \sum_{\delta=1}^{W}|\boldsymbol{r}_\delta(t+\Delta t) - \boldsymbol{r}_\delta(t)|/(W\Delta t)$ for the vortices in the system, where $\boldsymbol{r}_\delta = (x_\delta, y_\delta)$ with $\delta = 1, 2, \cdots, W$ stands for the instantaneous position of each vortex core. As an example, we discuss the $15\xi_1 \times 15\xi_1$ mesoscopic sample in the presence of an isotropic impurity with the disorder strength $g = -0.5$ here. In the procedure of simulations, we notice that

the vortex number in the sample will no longer change beyond $t \approx 10^3 t_0$. We then plot the variations of $\bar{v}$ with $t$ for the vortex cluster state and the vortex lattice phase in Fig. 14. It can be seen from Fig. 14 that the $\bar{v}$ evolves with $t$ and eventually stabilizes at $t < 10^4 t_0$. Therefore, it is justified for us to take the snapshots at $t = 10^4 t_0$ to present the stable vortex dynamics.

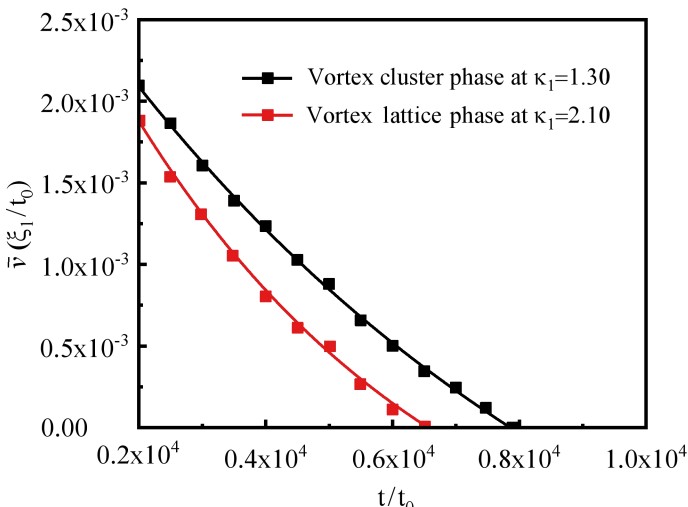

Figure 14: Variations of the average velocity $\bar{v}$ with time $t$ at the presence of an isotropic defect with the radius $0.5\xi_1$ in the $15\xi_1 \times 15\xi_1$ type-1.5 superconductor. We set the external magnetic field $H = 0.8H_0$ in the numerical simulations.

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
