# Peer review of "Impurity effect and vortex cluster phase in mesoscopic type-1.5 superconductors"

_SciPost Physics, doi:SciPost Phys. 19, 140 (2025)_

## Round 1 · Referee Report · Anonymous (Referee 1) · 2025-1-5

Strengths

This paper addresses a topic of significant interest: the interplay between \ defects and vortices, specifically in the context of two-component superconductors. While thousands of studies explore this relationship in single-component systems, very few have extended this analysis to two-component models. This is an important gap in the literature, especially given the growing number of materials experimentally identified as type-1.5 superconductors. These discoveries underscore the need for theoretical investigations into vortex structures in two-component systems under the influence of pinning. The strength of the paper is that it focuses on this underexplored yet promising research area.

Weaknesses

For a paper to be published in a good journal, it is required, in my opinion, to present a more systematic study.

Report

While the paper addresses an important and underexplored topic, it lacks the systematic analysis required for publication in a top-tier journal. A more comprehensive study could include investigations of various defect configurations and models of disorder. For instance, vortex clustering can arise not only from attractive intervortex interactions but also as a consequence of pinning or disorder. The authors could explore whether there exists a critical disorder strength, or density of pinning centers, where the role of attractive interactions becomes particularly significant.

Additionally, a comparative analysis of type-II and type-1.5 vortex systems under different disorder models would add valuable context and broader relevance to the study. The inclusion of both correlated and uncorrelated disorder is also interesting.

With the current material, the paper is suitable for publication in some journal, but not yet at the level of a leading one. A more systematic approach, as outlined above, would strengthen its impact and contribution to the field.

The authors state: "As we know, each condensate in two-band superconductors is predicted to support vortex excitation with fractional quantum flux [4, 5]." However, this statement is problematic. Reference [4] does not actually consider fractional vortices; instead, it deals with an infinitely thin loop. For such a configuration, the enclosed flux is effectively zero. The misinterpretation in [4] arises from the assumption that the phase winding can differ from an integer multiple of 2𝜋, which is incorrect.

Additionally, the authors should update their discussion to acknowledge recent experimental advances. A Science paper published in 2023 and two arXiv preprints from 2024 report experimental observations of fractional vortices.

Recommendation

Ask for major revision

  • validity: -
  • significance: -
  • originality: -
  • clarity: -
  • formatting: -
  • grammar: -

Author:  Wang Guo  on 2025-07-24  [id 5673]

(in reply to Report 1 on 2025-01-05)
Category:
answer to question

We would like to thank the anonymous referee for very inspiring suggestions. We have provided the point-by-point response to comments from Reviewer 1, see file attachment "Type-1.5-response-1.pdf" for detail.

Attachment:

Type-1.5-response-1.pdf

---

## Round 1 · Referee Report · Anonymous (Referee 2) · 2025-1-9

Report

The paper studies two-band superconductors in the presence of impurities in two dimensions. The authors numerically solve time-dependent Ginzburg-Landau equations for two-band superconductors to obtain stationary solutions. They analyze the magnetic flux and superconducting order parameter distributions for one isotropic, two isotropic, and one non-isotropic impurity with $C_4$ symmetry in the presence of the external magnetic field. For each case, the authors demonstrate solutions corresponding to type-1, type-1.5, and type-2 superconductors. In the second regime, the authors observe vortex clustering around the impurities, whose pattern is different for each of the three impurity types.

The paper contains a detailed description of the numerical algorithm and provides enough information to reproduce the demonstrated results. However, the authors do not fully utilize their opportunity to investigate the properties of their system.
First of all, the role of the impurities was not thoroughly analyzed. To improve this, the authors may, for example, consider how impurities and their strength affect the transitions between the Meissner, vortex cluster, and vortex lattice phases.
Secondly, the authors do not discuss how they can distinguish effects caused by impurities and boundaries. I suggest the authors compare the results for various system sizes and aspect ratios to separate the boundary effects.

In addition to this, there are several minor points that need to be clarified.
The choice of the value for the parameter $t$ (lines 153, 167, 175) requires some motivation.
The authors call the phase, demonstrated in Figs. 1a, 2a, 3a, a "Meissner phase". However, there is a nonzero magnetic field density on the impurities in this phase. This contradiction should be clarified.

Recommendation

Ask for major revision

  • validity: -
  • significance: -
  • originality: -
  • clarity: -
  • formatting: -
  • grammar: -

Author:  Wang Guo  on 2025-07-24  [id 5674]

(in reply to Report 2 on 2025-01-09)
Category:
answer to question

We would like to thank the anonymous referee for very inspiring suggestions. We have provided the point-by-point response to comments from Reviewer 2, see file attachment "Type-1.5-response-2.pdf" for detail.

Attachment:

Type-1.5-response-2.pdf

---

## Round 1 · Referee Report · Anonymous (Referee 3) · 2025-1-28

Strengths

1- clearly written 2-easily reproducible 3-focused on a simple model

Weaknesses

1-mathematically naive 2- not very comprehensive

Report

This paper presents a numerical study of vortex solutions of a simple two-component Ginzburg Landau theory in a rectangular domain with prescribed impurities. Energy minimizers are found by solving the gradient flow equation for the energy functional for large times. An advantage of this method is that the PDEs can be formulated in a standard computational package (COMSOL) implementing a finite element method. This is not the first paper to take this approach, but still, going through the formulation in detail is potentially useful for the condensed matter theory community. The main advantage of finite elements (ability to accommodate irregular domains) is not relevant here since the work considers only rectangles. A disadvantage of the numerical approach is that gradient flow is extremely slow to converge - conjugate gradient methods or arrested Newton flow are many times faster.

The focus of the paper is unclear: are we studying impurities, or boundary effects? It would be informative to compare the results on mesoscopic domains with no impurities and varying sizes and shapes, and on large domains with impurities. Restricting the study to the simplest TCGL model is justified I think: the parameter space of the full model is too large to be surveyed.

Since boundary effects are important for the results here, I'm a bit troubled by the authors' choice of boundary condition. It's true that the conditions assumed ensure that no supercurrent passes through the boundary, but they are much stronger than is required by that condition. They are not gauge invariant, and they impose that each "component" of the supercurrent (associated with each condensate) is confined separately. For the simple model studied here, which has U(1) x U(1) symmetry, and hence separately conserved supercurrents, this may well be justified, but for two component GL models in general this strikes me as a very artificial assumption. The authors justify their choice by citation to the literature, but having followed the thread back 3 links I still haven't found a derivation of them. Given the importance (presumably) of boundary effects, I think a derivation of the boundary conditions from physical/mathematical principles is needed.

The figures refer to "evolution" of physical quantities. This is misleading as it is unrelated to the "time evolution" used to generate the solutions.

Requested changes

1- derive the boundary conditions from first principles 2- extend the numerical investigation to separte out boundary effects and impurity effects. 3- clarify the meaning of "evolution"

Recommendation

Ask for major revision

  • validity: ok
  • significance: ok
  • originality: ok
  • clarity: good
  • formatting: good
  • grammar: good

Author:  Wang Guo  on 2025-07-24  [id 5675]

(in reply to Report 3 on 2025-01-28)
Category:
answer to question

We would like to thank the anonymous referee for very inspiring suggestions. We have provided the point-by-point response to comments from Reviewer 3, see file attachment "Type-1.5-response-3.pdf" for detail.

Attachment:

Type-1.5-response-3.pdf

---

## Round 2 · Referee Report · Anonymous (Referee 2) · 2025-8-26

Report

The authors significantly improved the manuscript, providing the requested results for systems with different sizes and impurity strengths. In the updated text, the $L-\kappa_1$ phase diagram shows how the vortex cluster formation depends on the system size in the absence of impurity. The $g-\kappa_1$ phase diagram demonstrates how the presence of the impurities and their strength modify the cluster formation.
However, it should be clarified how these diagrams were obtained, and, specifically, which criteria were used to distinguish the vortex lattice and cluster phases. The studied systems have only 10-15 vortices, and the authors should motivate how they distinguish the phases while working with such a small number of vortices.
In addition to that, the current version of the text does not provide any details about the method used to draw the phase separation lines. I recommend that the authors specify which points they check in the parameter space and explain how they extrapolate the results.

Finally, I kindly ask the authors to increase the image resolution and the font size for Fig. 2 and Figs. 4-11.

Requested changes

1 - formally specify the difference between the vortex lattice phase and the vortex cluster phase
2 - describe the method used for the calculation of the phase diagrams
3 - increase the image resolution and the font size for Fig. 2 and Figs. 4-11

Recommendation

Ask for minor revision

  • validity: ok
  • significance: ok
  • originality: ok
  • clarity: good
  • formatting: good
  • grammar: -

Author:  Wang Guo  on 2025-10-27  [id 5952]

(in reply to Report 1 on 2025-08-26)
Category:
answer to question

We would like to thank the anonymous referee for very helpful suggestions. We have provided the point-by-point response to comments from Reviewer 2, see file attachment "Type-1.5-response-2.pdf" for detail.

Attachment:

Type-1.5-response-2.pdf

---

## Round 2 · Referee Report · Anonymous (Referee 1) · 2025-9-20

Strengths

While the morphology of vortex clusters in clean type-1.5 superconductors was studied in many works, the effects of pinning and defects have not been well investigated. This study explored a subject that is experimentally relevant and timely. It is especially timely because of the 2023 Science paper by Kathryn Molel and Yusuke Iguchi as well as works from Hong Ding's and Yihua Wang's groups that explicitly demonstrated the composite nature of vortices in multiband systems, directly related to the type-1.5 concept.

Weaknesses

In a long-term perspective, it would be interesting to expand that study and explore the effects as a function of interband Josephson coupling. Also, it would be interesting to see the modelling of larger-scale disorder, but that is beyond the scope of the current work.

Report

The authors study the effects in magnetically-coupled condensates. While other couplings are also interesting, that is the most generic coupling, which makes this study a clear exploration of the well-defined problem; many of the results will also qualitatively apply to the cases where fractional vortices are confined not only electromagnetically but also by Josephson coupling. Hence I recommend acceptance of this study.

Recommendation

Publish (meets expectations and criteria for this Journal)

  • validity: high
  • significance: good
  • originality: high
  • clarity: good
  • formatting: good
  • grammar: good

Author:  Wang Guo  on 2025-10-27  [id 5951]

(in reply to Report 2 on 2025-09-20)
Category:
answer to question
suggestion for further work

We would like to thank the editor and the anonymous referee for the comments and suggestions on the manuscript. We would also like to appreciate the reviewer's recognition of our research work and the decision to recommend our manuscript for publication. According to the referee's suggestion, we will investigate the effects of interband Josephson coupling and try to develop a larger-scale disorder model for two-band mesoscopic superconductors in the future studies.

---

## Round 2 · List of Changes

We would like to thank the editor and anonymous referees for very inspiring suggestions. We have provided the point-by-point responses to comments from the reviewers, see file attachments "Type-1.5-response-1.pdf", "Type-1.5-response-2.pdf" and "Type-1.5-response-3.pdf" for detail.

---

## Round 3 · Referee Report · Anonymous (Referee 2) · 2025-11-3

Report

The authors extended the manuscript by including the details of the procedure for obtaining the phase diagrams. The described method for defining the phase transition between the vortex cluster and vortex lattice phases is particularly interesting because it can be applied to mesoscopic systems with a relatively small number of vortices. The method allows for the construction of phase diagrams for various system sizes and disorder strengths, illustrating how disorder affects vortex cluster formation in finite systems. The final version of the manuscript provides a complete description of the numerical simulations performed and meets the general acceptance criteria of the journal.

Recommendation

Publish (meets expectations and criteria for this Journal)

---

## Editorial Decision

published